# Bull Sperm Capacitation Is Accompanied by Redox Modifications of Proteins

**DOI:** 10.3390/ijms22157903

**Published:** 2021-07-23

**Authors:** Agnieszka Mostek, Anna Janta, Anna Majewska, Andrzej Ciereszko

**Affiliations:** Department of Gamete and Embryo Biology, Institute of Animal Reproduction and Food Research of Polish Academy of Sciences, 10-748 Olsztyn, Poland; a.janta@pan.olsztyn.pl (A.J.); a.majewska@pan.olsztyn.pl (A.M.); a.ciereszko@pan.olsztyn.pl (A.C.)

**Keywords:** sperm capacitation, *Bos taurus*, redox proteomics, reversible oxPTMs

## Abstract

The ability to fertilise an egg is acquired by the mammalian sperm during the complex biochemical process called capacitation. Capacitation is accompanied by the production of reactive oxygen species (ROS), but the mechanism of redox regulation during capacitation has not been elucidated. This study aimed to verify whether capacitation coincides with reversible oxidative post-translational modifications of proteins (oxPTMs). Flow cytometry, fluorescence microscopy and Western blot analyses were used to verify the sperm capacitation process. A fluorescent gel-based redox proteomic approach allowed us to observe changes in the level of reversible oxPTMs manifested by the reduction or oxidation of susceptible cysteines in sperm proteins. Sperm capacitation was accompanied with redox modifications of 48 protein spots corresponding to 22 proteins involved in the production of ROS (SOD, DLD), playing a role in downstream redox signal transfer (GAPDHS and GST) related to the cAMP/PKA pathway (ROPN1L, SPA17), acrosome exocytosis (ACRB, sperm acrosome associated protein 9, IZUMO4), actin polymerisation (CAPZB) and hyperactivation (TUBB4B, TUB1A). The results demonstrated that sperm capacitation is accompanied by altered levels of oxPTMs of a group of redox responsive proteins, filling gaps in our knowledge concerning sperm capacitation.

## 1. Introduction

Mammalian sperm capacitation is a complex process that occurs in the oviduct and is essential for the fertilisation of a mature oocyte. The successful completion of capacitation, which involves many biochemical and morphological changes, enables the spermatozoon to bind to the zona pellucida, penetrate it and fuse with the oolemma. Biochemical changes in the plasma membrane and other subcellular compartments have been associated with sperm capacitation [1]. At the early stages of capacitation, a series of cellular events occurs. These include, among others, the production of cAMP by the adenylyl cyclase (AC), the activation of calcium channels, reactive oxygen species (ROS) generation, cholesterol efflux from the plasma membrane, the alkalinisation of sperm plasma and the activation of protein kinases [2].

It is well known that mammalian sperm capacitation is an oxidative process [3,4]. The production of ROS is an early event during the series of modifications that occur then, and a variety of ROS species are produced while spermatozoa capacitate [3,5]. ROS may be produced in different sperm cell compartments; however, it remains unknown which source exactly is responsible for generating ROS that are involved in capacitation [6]. Growing evidence suggests that oxidases localised in the sperm plasma membrane, such as nitric oxide synthase (NOS) or oxidase with intrinsic NOS activity, are the main source of ROS and reactive nitrogen species (RNS) in bull and human spermatozoa [3]. ROS and RNS are formed by the action of O_2_^•−^ and redox enzymes. Although a small percentage of H_2_O_2_, one of the most abundant ROS, may be generated spontaneously, the majority is formed as a result of O_2_^•−^ dismutation catalysed by superoxide dismutases (SODs) and other oxidases with dismutase activity. Most OH^•^, on the other hand, is produced from H_2_O_2_ and O_2_^•−^ in the reaction catalysed by iron or copper ions [7]. In turn, ONOO^-^ is formed in the reaction between NO and O_2_^•−^. ROS generation causes the oxidation of cholesterol to oxysterols, which results in a dramatic increase in membrane fluidity. The combined actions of O_2_^•−^, HCO_3_^−^ and Ca^2+^ activate soluble adenylyl cyclase (sAC), which stimulates cAMP production and the further activation of protein kinase A (PKA). In all species studied to date, including bull, this reaction leads to tyrosine phosphorylation [6].

Sperm mitochondria are an important source of O_2_^•−^, generating low levels of ROS during steady-state respiration. Given that mitochondria are extremely susceptible to electron leakage and O_2_^•−^ easily passes through the membrane via voltage-dependent anion channels, mitochondrial ROS can be extremely damaging to the cell [6]. A major, often overlooked, source of ROS is dead or damaged sperm cells with impaired mitochondria [8]. The excessive accumulation of reactive oxygen and nitrogen species may lead to oxidative stress, especially when this is accompanied by insufficient activity of ROS scavenging systems [9]. On the other hand, the precisely localised and controlled synthesis of ROS and RNS in cells can be beneficial in certain physiological states. Various reversible oxidative modifications, known as redox signalling, are triggered by ROS and RNS and are recognised as an important element of the regulatory mechanism of cell metabolism [10,11].

A key role in redox signalling is played by highly reactive protein thiol groups on cysteine residues. They easily react with ROS and RNS and form either reversible or irreversible oxidative post-translational modifications (oxPTMs) which have a substantial impact on protein function [12]. Reversible oxPTMs of cysteines, such as the formation of disulphide bridges, S-nitrosylation, sulphenylation and S-glutathionylation, can be transformed into other stable oxidised groups or reduced back to form free thiols. Cysteine sulphonation can only be reduced back by sulphiredoxin or sestrin. In contrast, the irreversible covalent sulphinylation of cysteine residues in tryptophan, tyrosine, lysine and histidine may result in the loss of protein function due to improper folding, aggregation or degradation and often coincides with cellular damage [12]. It is assessed that around 10% of all the cysteine residues present in the cell proteome are susceptible to reversible redox modifications [13]. The information contained within the redox status of the protein, including the site, type and extent of the oxidative modification, is transduced through the cell to induce certain molecular responses. Redox signalling via reversible oxPTMs leads to alterations in redox-sensitive target proteins by affecting their activity, stability, localisation and interactions with other molecules. As such, the reversible oxPTMs may regulate a plethora of cellular processes under both physiological and pathological conditions [11]. It should be emphasised that the complexity of cell signal transduction is further increased by the cross-talk between redox signalling via reversible oxPTMs and phosphorylation-based signalling systems [14,15].

According to the current knowledge, sperm capacitation is accompanied by an increased concentration of ROS and RNS that triggers the redox signalling cascade. However, the mechanism of redox signalling during sperm capacitation remains unknown. Thus far, the involvement of a small group of redox-regulated proteins, including PKC, Ras and AC, has been shown [3]. Our research was established based on two facts: reversible oxPTMs are a hallmark of redox signalling, and redox modifications of proteins can be an integral mechanism of capacitation. Therefore, in our study, we aimed to investigate whether bull sperm capacitation is accompanied by reversible oxPTMs of proteins, which may indicate a redox signal transmission.

## 2. Results

### 2.1. Evaluation of Sperm Capacitation: CTC Staining, LPC-Induced Acrosome Reaction and Tyrosine Phosphorylation Level 

A constant increase in the fraction of capacitated spermatozoa was observed over time in capacitation-supporting buffer as chlortetracycline (CTC)-stained pattern B (Figure 1, Figure 2A). A significant difference was observed after 2 h of incubation, and the largest accumulation of pattern B among different timepoints was observed after 6 h of capacitation, reaching 31 ± 8.1% (Figure 2A). The number of sperm showing a spontaneous acrosomal reaction increased with the duration of capacitation. The levels of acrosome reaction induced by lysophosphatidylcholine (LPC), which is a determinant of capacitation, increased significantly after 4 h and further after 6 h of incubation, finally reaching 53 ± 4.4% (Figure 2B). The incubation of the bull sperm under conditions supporting capacitation also increased the level of tyrosine phosphorylation. After 6 h of capacitation, the tyrosine phosphorylation of sperm proteins increased by 45% in relation to the 0 h time (Figure 3).

### 2.2. Sperm Motility and ROS-Positive Cell Content during Sperm Capacitation

The incubation of spermatozoa in capacitation-supporting buffer for 2, 4 and 6 h caused the head-to-head agglutination of spermatozoa (Figure 4), which prevented the correct measurement using the CASA system. Therefore, after these incubation times, the total sperm motility was assessed subjectively. The percentage of motile sperm decreased between 0, 2 and 4 h of capacitation without a further decrease between 4 and 6 h (Figure 5A). The analysis of movement trajectory parameters was possible only at 0 h of capacitation. The values of sperm movement trajectory parameters at 0 h of capacitation were as follows: ALH—amplitude of lateral head displacement: 7.6 ± 0.65; VCL—curvilinear velocity: 204 ± 6.4; VSL—straight line velocity: 120.3 ± 5.9; BCF—beat cross frequency: 41.2 ± 0.93; LIN—linearity: 61.2 ±3.4. 

The percentage of ROS-positive (ROS+) spermatozoa initially decreased between 0 and 2 h of capacitation. The increase of ROS+ spermatozoa was observed between 2 and 4 h of capacitation and further between 2 and 6 h, reaching 30.5 ± 5.7 % (Figure 5B).

### 2.3. Overall Level of Reversible Oxidative Post-Translational Modifications in Proteins (oxPTMs)

The overall level of protein-reversible oxPTMs did not change during the first 2 h of incubation under capacitation-supporting conditions and increased between 2 and 4 h and further between 4 and 6 h (Figure 6A,B). The highest level of protein-reversible oxPTMs was detected after 6 h of incubation under capacitation-supporting conditions, which resulted in the selection of times of 0 and 6 h for the further evaluation of oxPTMs by two-dimensional gel electrophoresis (2D PAGE).

### 2.4. Identification of Redox-Modified Proteins

Two-dimensional gel electrophoresis showed 48 protein spots corresponding to 22 proteins with altered levels of oxPTMs (*p* < 0.05, fold change >2) (Figure 7). Among the identified protein spots, 39 showed an increase in the level of oxPTMs, while 9 protein spots showed a decrease in oxPTMs after 6 h of capacitation (Figure 7, Table 1). STRING protein association analysis showed multiple potential protein–protein interactions of the identified redox-modified proteins (Figure 8). Results obtained from the gene ontology analysis of biological processes showed that the identified proteins with changed oxPTM levels are related to capacitation, glycolysis, NADH regeneration, mitochondrial ATP synthesis-coupled proton transport and pyruvate metabolism.

## 3. Discussion

### 3.1. Research Assumptions and Rationale for Experimental Design

In our study, we aimed to investigate whether bull sperm capacitation is accompanied by reversible protein oxPTMs, which may indicate redox signal transduction. In our experimental system, bicarbonate and heparin were used as capacitation inducers. Although the most common sperm capacitation buffers also contain bovine serum albumin (BSA), we did not include this in the incubation medium as it has been shown to interfere with cytometric analyses [16,17,18], which was also confirmed by our preliminary research (data not shown). Earlier studies showed that bicarbonate and heparin effectively trigger capacitation without other inducers in bovine sperm [18,19]. In our approach, the sperm capacitation process was monitored using three independent methods (CTC staining, LPC-induced acrosome reaction and tyrosine phosphorylation assessment). A gel-based proteomic approach allowed for the observation of changes in the level of reversible oxPTMs manifested by the reduction or oxidation of susceptible cysteines in proteins. This study demonstrated that bull sperm capacitation is indeed accompanied by a change in the level of oxPTMs of a specific group of proteins, which are likely to be key players in redox signalling mediating sperm capacitation.

### 3.2. Bull Sperm Capacitation Verification

Ejaculated spermatozoa must undergo a complex and time-dependent process, collectively called capacitation, that involves numerous biochemical and physiological changes, such as intracellular calcium influx or increased plasma membrane fluidity [20]. The successful completion of capacitation is mandatory for the spermatozoa to fuse with the zona pellucida surrounding the oocyte and release the proteolytic enzymes during the exocytotic event called the acrosome reaction [21,22]. The three methods of capacitation evaluation used in this study confirmed that the incubation of spermatozoa in capacitation-supporting conditions caused biochemical changes characteristic for capacitated sperm, including the increased susceptibility of spermatozoa to undergo LPC-induced acrosome reaction, an increase in intracellular calcium level (CTC staining) and elevated protein tyrosine phosphorylation. Moreover, head-to-head sperm agglutination, which increased over time under capacitation-supporting conditions, also indicated the capacitated status of the sperm. In earlier reports, it was shown that cAMP-induced sperm agglutination is correlated with cytoplasmic free calcium accumulation [23]. 

### 3.3. ROS Production during Sperm Capacitation 

Capacitation is known to be a redox-regulated process, but the exact signalling mechanism is poorly understood. The production of the controlled levels of ROS by the spermatozoon is one of the key features of sperm capacitation since ROS act as secondary messengers involved in various signal transduction pathways. This phenomenon, which is essential for acquiring fertilisation ability, was first reported in human [4], bovine [24] and equine spermatozoa [25]. Our results showed that the percentage of ROS-positive sperm increased during capacitation, but ROS production was at a relatively low level throughout the incubation period, not exceeding 31% after 6 h (Figure 5B), which is consistent with previous reports on the generation of small amounts of ROS required for proper redox signalling during sperm capacitation [24,26]. Spermatozoa generate ROS for a limited period of time: after their release from the male reproductive tract to the time of their ascent through the female oviduct to the site of fertilisation. In the case of fertilisation failure, the continuous generation of ROS in the sperm cell will lead to a state of over-oxidation. Ultimately, it may cause the sperm to initiate apoptosis [8].

### 3.4. Redox Modification of SOD and DLD during Sperm Capacitation 

In our experiment, superoxide dismutase (SOD) was identified in two proteoforms as being redox modified. Superoxide dismutase catalyses the dismutation of the superoxide anion free radical (O_2_^−^) into molecular oxygen and hydrogen peroxide (H_2_O_2_). Although there is no information of oxPTMs of SOD during capacitation, redox modifications of this enzyme are one of the best-documented oxPTMs in somatic cells [27]. The oxidation of SOD Cys111 causes the misfolding of the protein and promotes monomer formation, which is the initiating step for SOD1 aggregation [27,28]. After 6 h of capacitation, SOD underwent a reduction of Cys residues, which may indicate the transformation of SOD into a more stable form, enabling more efficient catalytic functioning. Existing evidence shows that SOD activity is sufficient to account for all of the H_2_O_2_, which is the most important redox signalling messenger during sperm capacitation [29]. In the process of H_2_O_2_-dependent signal transduction, H_2_O_2_ selectively targets specific redox-sensitive proteins containing critical thiols. The oxidation signal is transferred to their downstream signalling proteins through the consecutive thiol-disulphide exchange reactions [30]. Taken together, our results indicate that Cys reduction of the SOD enzyme takes place during bull sperm capacitation, likely resulting in increased SOD stability, which is needed to generate H_2_O_2_—the most important redox messenger. 

Dihydrolipoyl dehydrogenase (DLD), which was recorded as oxidised under capacitation-supporting conditions, is an important enzyme involved in ROS generation. DLD is part of the pyruvate dehydrogenase multienzyme complex (PDHc), which has the ability to reduce O_2_ to O_2_^−^ or ferric to ferrous iron, which then catalyses the production of hydroxyl radicals [31,32]. Due the reactive disulphide bridge in its structure, DLD is susceptible to reversible oxPTMs [33]. This enzyme exhibits capacitation-dependent tyrosine phosphorylation in hamster spermatozoa [34], and the inhibition of its activity causes a significant decrease of hyperactivation and the acrosome reaction [35]. The significance of DLD for sperm capacitation is probably associated with its ability to produce ROS and cAMP, as their synthesis is significantly limited in the presence of DLD inhibitors. It was concluded that cAMP and calcium play a role downstream of DLD during hamster sperm capacitation [35]. Until now, sperm membrane NADPH oxidases were considered as the main sources of ROS in capacitation. Considering the redox modification of DLD during capacitation, coupled with the pro-oxidative properties and oxPTM susceptibility, DLD can be a novel redox-regulated candidate responsible for ROS production during sperm capacitation.

### 3.5. GST and GAPDH as Potential Redox Signal Transducers Involved in Capacitation

To a small extent, the redox signal can be transmitted to sensitive thiols directly from H_2_O_2_; however, the vast majority of evidence points to the involvement of specific proteins that function as redox signal transducers. Such redox transducers are characterised by high reactivity with H_2_O_2_ and the presence of reactive Cys in their active sites [36]. Under capacitation-supporting conditions, two enzymes exhibiting redox signal transducer potential were identified: glutathione S-transferase (GST) and testis-specific glyceraldehyde-3-phosphate dehydrogenase (GAPDH). Both enzymes are highly reactive with H_2_O_2_ and possess conserved Cys in their active site [36], which makes them prone to being central redox signal transducers. The functions and activities of target (redox responsive) proteins rely on the formation of oxPTMs, such as disulphide formation (S-S), S-glutathionylation (SSG), S-nitrosylation (SNO) and S-sulphenylation (SOH). Glutathione S-transferase, which was identified in our study, catalyses the S-glutathionylation of target protein cysteines and thus is a key enzyme in redox signalling [37]. The role of GST in regulating the forward reaction of protein S-glutathionylation has already been described [38]. It was shown that the function of GST relies on the auto-S-glutathionylation of two residues (Cys47 and Cys101), which affects its catalytic activity and binding to target proteins. Moreover, the current knowledge indicates a significant relationship between GST and male fertility and its important role not only in detoxification but also in the regulation of cellular signalling. It was hypothesised that sperm GSTs play a role in sperm capacitation via MAPK molecular signalling pathways [39]. The redox regulation of GST via Cys oxidation, reported in our study under capacitation-supporting conditions, indicates that GST may be involved in the formation of oxPTMs on target proteins. Furthermore, these results indicate that, among the different oxPTMs, GST-dependent S-glutathionylation may be a dominant protein redox modification during capacitation. 

Additionally, GAPDH involvement seems to be particularly important in redox signalling during capacitation since as many as eight proteoforms with increased levels of oxPTMs were identified after 6 h of capacitation. The current knowledge confirms that GAPDH can be a potential central redox mediator in cells [40]. GAPDH is known not only as a key glycolytic enzyme but also as a protein that has versatile functions in other biological processes. Catalytic Cys of GAPDH is subjected to diverse reversible oxPTMs, inhibiting glycolytic activity and promoting the pleiotropic functions of GAPDH. These include microtubule cytoskeleton organisation, involvement in apoptotic processes and membrane trafficking [40,41]. H_2_O_2_-dependent oxidation has been shown to cause changes to the 3D structure of GAPDH by linking Cys152 and Cys156 via disulphide bridging [42] and disrupting its cytosolic (glycolytic) activity. Interestingly, this discovery suggests that GAPDH is a potential mediator and translator of the oxidation state of the cell [40]. The eight proteoforms identified in our research belong to the bovine testis-specific subtype of GAPDH characterised by a unique N-terminal proline-rich extension of 61 amino acids. In human sperm, this proline-rich extension enables the binding of GAPDH to the fibrous sheath of the spermatid flagellum, confirming the importance of this enzyme for sperm motility [40]. It has been demonstrated that mice spermatozoa with GAPDH gene knockout produce decreased levels of ATP and are characterised by impaired fertility [43]. Notably, the testis-specific subtype of GAPDH, identified in our study, possessed two additional cysteine residues compared to somatic variants, increasing the ability to conduct redox signal transduction.

In addition to various redox Cys modifications, GAPDH is prone to undergo numerous other non-redox PTMs, including phosphorylation, acetylation and glycosylation. The described PTMs may significantly change the properties of GAPDH and the functions performed [40]. Since GAPDH has been identified in our research in multiple proteoforms, it can be assumed that the identified redox-regulated proteoforms might be simultaneously subjected to other PTMs. The properties of GAPDH as a mediator of the redox signal transmitter indicate that bull sperm GAPDH can play an important role in transferring the oxidation signal to the downstream redox-sensitive target proteins. 

### 3.6. Redox Regulation of Proteins Related to the PKA Signalling Pathway 

Some of the redox-regulated proteins identified in our study are known to be the components of previously recognised capacitation signalling pathways such as the cAMP/PKA pathway. As an early event, superoxide and nitric oxide activate adenyl cyclase (AC) which catalyses the production of cAMP during sperm capacitation [4,44]. This event activates protein kinase A (PKA), which is essential for supporting tyrosine phosphorylation—a hallmark of sperm capacitation in all species studied to date [26]. PKA localises to specific cellular structures and organelles by binding to AKAP (A-kinase anchoring protein), which in turn is regulated by two proteins identified in our study as redox-modified, namely ROPN1L (ROPN1-like protein) and SPA17 [45]. 

Presumably, the redox regulation of proteins controlling the activity of AKAPs during capacitation may lead to the enhancement of AKAP activity, consisting of the coordination of the action of several signalling molecules by combining cAMP production, kinases, phosphatases and their substrates [46]. The correct redox modification of the AKAP regulatory proteins ROPN1L and SPA17 may be essential for the proper phosphorylation of proteins during capacitation, considering previous studies showing that, in the mice sperm lacking ROPN1L, there was a reduction in the cAMP-dependent protein kinase (PKA) phosphorylation of FSCB (fibrous sheath CABYR binding protein) and in capacitation-induced tyrosine phosphorylation [45]. Overall, it is possible that redox modifications of proteins during capacitation are a part of a more complex signalling system, and redox-regulated target proteins, such as ROPN1L and SPA17, are associated with other important phosphorylation-based signalling pathways. The exact mechanisms of AKAPs through the redox modification of ROPN1L and SPA17 should be elucidated in further studies.

### 3.7. Redox-Regulated Proteins Related to Acrosome Exocytosis during the Acrosome Reaction

The series of cellular and molecular changes that takes place during capacitation ultimately enables sperm acrosome exocytosis and interaction with the oocyte. The obtained results indicate that redox modifications of some acrosome proteins may accompany the known biochemical changes preceding the acrosome reaction. Under capacitation-supporting conditions, acrosin-binding protein (ACRBP) was redox-regulated via the reduction of cysteine. ACRBP is an enzyme that regulates the proteolytic disassembly of the acrosomal matrix. The ACRBP-dependent activation of acrosin protein is required during acrosomal exocytosis for the efficient dispersion of acrosomal content. Otherwise, the process is slower. Moreover, the previously reported capacitation-associated phosphorylation of this enzyme is possibly a preparatory step for the autoactivation of proacrosin to acrosin during acrosomal exocytosis, resulting in a disassembly of the acrosomal matrix [47].

Our study revealed that sperm acrosome-associated protein 9 is also redox-regulated via reduction after 6 h of capacitation. Although the role of sperm acrosome-associated protein 9 is not fully understood, this protein forms a complex with acrosomal calcium-binding proteins calreticulin and caldendrin [48]. Therefore, it is possible that sperm acrosome-associated protein 9 is a novel player in the regulation of AR via calcium-mediated signal transduction.

Six redox-modified IZUMO 4 proteoforms were identified in our study, three of which were reduced, and another three were oxidised in capacitated sperm. IZUMO 4 is a member of the IZUMO multiprotein family containing four isoforms of IZUMO, of which 1, 2 and 3 are transmembrane proteins and 4 is a soluble, intra-acrosomal protein involved in the formation of protein complexes [49]. In porcine spermatozoa, the functional exposure of IZUMO proteins is essential for gamete fusion [50]. The presence of multiple proteoforms of IZUMO 4 in our study may indicate various post-translational modifications of this protein (e.g., phosphorylation, acetylation, glycosylation, ubiquitylation) [51]. Since different proteoforms of the IZUMO 4 protein were differentially redox-modified, the interaction of oxPTMs likely occurs with other post-translational protein modifications. It is worth mentioning that various proteoforms may have different susceptibilities to oxidation and other proteomic alterations. It seems that redox regulation concerns specific acrosomal proteins related directly to the exocytosis of acrosome content, acrosin activation and gamete fusion.

### 3.8. Redox-Regulated Cytoskeletal Proteins Related to Actin Polymerisation and Hyperactivated Motility during Capacitation 

Two proteoforms of F-actin capping proteins (CAPZB) involved in actin polymerisation were found to be oxidised during capacitation. In the course of capacitation, the globular (G)-actin proteins polymerise to form a filamentous (F)-actin. The process depends on the activation of PKA and phospholipase D, as well as on protein tyrosine phosphorylation, and enables the translocation of phospholipase C from cytosol to the plasma membrane. In turn, the depolymerisation of F-actin is performed by the actin-severing proteins upon the binding of the capacitated sperm to the zona pellucida. The disintegration of actin fibres precedes the acrosome reaction and the fusion of sperm plasma and acrosomal membranes with the oocyte [52]. CAPZB is highly involved in actin polymerisation during sperm capacitation via the regulation of the access to the free barbed ends of actin filaments. In mouse sperm, CAPZB is present in the anterior acrosome before capacitation and then is redistributed to the post-acrosomal compartment, finally diminishing when the acrosome reaction is completed [53]. Based on the obtained results, it is likely that CAPZB redox regulation is involved in the actin polymerisation mechanism during capacitation and/or depolymerisation during the acrosome reaction as an additional mechanism accompanying PKA activation, protein tyrosine phosphorylation and phospholipase D activation.

In addition to the proteins related to actin polymerisation, the oxidation of two β-tubulin proteoforms (TUBB4B) and one α-tubulin proteoform (TUB1A) was recorded during capacitation. Tubulins form microtubules in the axoneme and are a fundamental part of the sperm movement apparatus. Capacitation is often accompanied by the change in sperm motility pattern called hyperactivation, characterised by a distinct figure-of-eight tail whip and less progressive movement manner, which are likely to prevent the binding of the sperm to the epithelial cells [54]. The redox modification of TUBB4B and TUB1A through oxPTMs may be associated with sperm hyperactivation. Several mechanisms involved in the hyperactivation of sperm motility have been discovered so far, including the Ca^2+^ and cAMP regulation of microtubule sliding [55], CATSPER channel functioning [56], flagellar protein phosphorylation and dynein activity inhibition [57]. To our knowledge, however, redox regulation has not yet been mentioned. Moreover, hyperactivated motility in sperm capacitation was found to be mediated by phospholipase D-dependent actin polymerisation [58]. Therefore, the redox modifications of the cytoskeletal proteins CAPZB, TUBB4B and TUB1A may be closely related to each other. It seems that redox modifications of proteins during sperm capacitation are pleiotropic, affecting many mechanisms directly related to capacitation, acrosome reaction (such as actin polarisation/depolarisation) and movement hyperactivation at the same time. 

### 3.9. Redox-Regulated Proteins Involved in Energy Metabolism during Capacitation

Our research revealed a surprisingly large amount of redox-modified proteins related to energy metabolism, including glycolysis, the TCA cycle and oxidative phosphorylation (OXPHOS). Among the glycolytic enzymes, the oxidation of four proteoforms of ENO and three proteoforms of ALDOA was observed. Moreover, the oxidation of enzymes involved in the TCA cycle was recorded: CS (two proteoforms) and ME2 (three proteoforms). Finally, the oxidation of key proteins for OXPHOS was identified: ATP synthase subunits α (four proteoforms), β and D and the cytochrome b-c1 complex subunit1 (UXCRC1). Physiologically, sperm in the epididymis remain dormant; however, after ejaculation, the instant energy production is essential to initiate sperm movement and capacitation [59]. Until recently, glycolysis was established to be important in supporting mouse sperm capacitation [60]. Conversely, more current studies have confirmed that capacitating sperm actively increase the uptake and utilisation of glucose and showed that both glycolysis and OXPHOS accelerate during mouse sperm capacitation. The increase in OXPHOS was shown to be dependent on glycolysis, providing evidence for a link between OXPHOS and glycolysis in sperm [59]. Although the elevated energy demand during mammalian sperm capacitation is well established, the exact mechanism of energy metabolism modification during capacitation remains unclear. Our results showing the redox regulation of several enzymes involved in the TCA cycle, OXPHOS and glycolysis during capacitation may be a partial answer to this issue. Our results showed that capacitation causes reversible oxPTMs in ENO, ALDOA, CS, ME2, ATP synthase and UQCRC1, which may trigger changes in energy metabolism during the capacitation of bull spermatozoa. The results obtained support the data that not only glycolysis but also TCA and OXPHOS can be important for energy metabolism during sperm capacitation [59].

## 4. Materials and Methods

Unless otherwise declared, all the reagents were purchased from the Sigma-Aldrich Chemical Company (St. Louis, MO, USA).

### 4.1. Research Material

Fresh semen was obtained from eight sexually mature Holstein Friesian bulls (*n* = 8). The ejaculates were collected by the employees of the Mazowieckie Center for Animal Breeding and Reproduction (Łowicz, Poland) in July and August 2020. The semen was diluted (1:1) in protein-free Bioxcel semen diluent (IMV Technologies, L’Aigle, France), stored at 4 °C and used within 3 h of harvesting.

### 4.2. Sperm Preparation and Concentration Measurement

Spermatozoa were separated from the diluent by centrifugation (300× *g* for 20 min). Subsequently, the remaining sperm pellet was washed three times with five-fold of the original volume of non-capacitating BO-SemenPrep medium (IVF Bioscience, Falmouth, UK) using a centrifuge (500× *g* for 5 min at RT). The Sperm Cell Counter NucleoCounter SP-100 (Chemometec, Allerød, Denmark) was used to measure sperm concentrations [61], which were further adjusted to 100 million/mL with a capacitation-supporting medium, as described in Section 4.3.

### 4.3. Sperm Capacitation

Sperm were incubated at 38 °C with 5% CO_2_ under capacitation-supporting conditions in capacitation medium compatible with flow cytometry developed by [18] with some modifications (100 mM NaCl, 3.1 mM KCl, 2 mM CaCl_2,_ 0.3 mM Na_2_HPO_4_, 21.6 mM Na-Lactate, 0.4 mM MgCl_2_ × 4H_2_O, 10 mM Hepes, 1 mM Na-pyruvate, 25 mM NaHCO_3_, 10 µg/mL heparin and 50 µg/mL gentamycin). Sperm were collected at 0, 2, 4 and 6 h, and three methods were used to evaluate capacitation levels: chlortetracycline (CTC) labelling, detection of the LPC-induced acrosome reaction and evaluation of tyrosine phosphorylation. 

### 4.4. Sperm Motility Measurement

The evaluation of sperm motility at 0 h was measured using computer-assisted sperm analysis (CASA) system as previously described [62]. Due to the capacitation-induced agglutination of spermatozoa, which prevents CASA measurement, the evaluations of sperm motility after 2, 4 and 6 h of capacitation were subjectively estimated by one operator based on the time-lapse photos generated by the CASA system.

### 4.5. Sperm Oxidative Status Evaluation

The oxidative status of bovine sperm was measured with a flow cytometry method including the CellROX Green reagent (Molecular Probes, Eugene, OR, USA). CellROX Green is a cell-permeable nuclear probe that is nonspecific to the type of free radical detected. When oxidised by intracellular free radicals, it binds to DNA and emits an intense green fluorescence. The aliquots containing 20 × 10^6^ cells/mL from samples incubated by 0, 2, 4 and 6 h under capacitation-supporting conditions (described in Section 4.3) were taken, and the CellROX Green reagent was added (6.25 μM). Cells were stained for 30 min at 38 °C in a 5% CO_2_ humidified atmosphere. As a negative control, 3 mM antioxidant N-acetylcysteine (NAC) was used. The positive control was obtained by using 2.5 mM menadione. The mentioned chemicals were added to separate aliquots of untreated sperm samples for 30 min prior to CellROX staining. Before flow cytometry, sperm preparations were diluted with PBS to obtain 300 cells/µL and placed in a 96-well plastic F-bottom clear plate (Greiner Bio-one, Frickenhausen, Germany). Samples were analysed with a GUAVA easyCyte 8HT Benchtop Flow Cytometer (Guava Technologies Inc., Luminex, Austin, TX, USA) with the InCyte guavaSoft 4.0 software for data acquisition and analysis. The fluorescence of CellROX Green-treated samples was excited with a blue 488 nm laser, and the emitted fluorescence was captured by means of a 515/30 nm GREEN-B filter set. The combination of forward scatter (FSC) and side scatter (SSC) signals were used to discriminate spermatozoa from debris, and 5000 cells were acquired for each analysis. GUAVA easyCYTE was calibrated daily with a Guava easyCheck Kit. The dividing point between ROS-positive (ROS+) and ROS-negative (ROS−) cell populations was determined using positive and negative controls (see above).

### 4.6. CTC Labelling of Bull Sperm

The proportion of capacitated spermatozoa was assessed under a fluorescent microscope using a chlortetracycline (CTC) assay. Upon entering the cell compartments containing free calcium, CTC becomes negatively charged and binds to calcium, which causes increased fluorescence. Bound CTC accumulates in the hydrophobic regions of a cell, resulting in staining patterns characteristic to the specific capacitation status of the sperm [63]. To assess the capacitation status of the spermatozoa, 30 µL of sample (100 × 10^6^/^mL^) was mixed by pipetting with the same volume of CTC solution (5.82 mM CTC, 5 mM cysteine, 130 mM NaCl, 20 mM Tris-HCl, pH 7.8) as described previously [64] with some modifications. The solution was prepared daily, protected from light, stored at 4 °C until required and brought to RT before addition to the sample. After 10 min of sperm incubation with CTC, cells were fixed by adding 4% paraformaldehyde in PBS (Santa Cruz Biotechnology, Dallas, TX, USA) to a final concentration of 1% and stored in the dark at 4 °C until evaluation. The evaluation was performed within 2 days from the experiment with no substantial fading of CTC fluorescence within this time. Ten microlitres of the sperm suspension was placed on a clean glass slide and covered with a cover slip. Cells were observed under an Axio Observer.Z1/7 fluorescence microscope (Carl Zeiss, Inc., Oberkochen, Germany) equipped with ZEN 2.3 blue edition software (Carl Zeiss). Cell count and detection was performed in the bright field mode (phase-contrast), and the CTC fluorescent signal was captured under 495/519 nm excitation/emission wavelengths. At least 200 spermatozoa per semen sample were counted and classified into three CTC staining patterns, as already reported [65]: uniform fluorescence over the entire sperm head (F pattern, non-capacitated sperm), fluorescence detected in the sperm head with a fluorescence-free band in the post-acrosomal region (B pattern, capacitated sperm) and low fluorescence signal over the entire sperm head, with a remaining positive signal along the equatorial segment (AR pattern, acrosome-reacted sperm) (Figure 1). Sperm with a nonspecific or intermediate fluorescent signal status were not selected for analysis.

### 4.7. Detection of the LPC-Induced Acrosome Reaction in Bull Sperm 

Another method to evaluate bull sperm capacitation status is the assessment of the lysophosphatidylcholine (LPC)-mediated acrosome reaction (AR). This method utilises the LPC feature to induce AR only in capacitated sperm [66]. To visualise sperm acrosomes, the lectin peanut agglutinin (PNA) Alexa Fluor 647 conjugate (Thermo Fisher Scientific, Waltham, MA, USA) was used. Thus, the high cell fluorescence observed with the flow cytometer indicates the presence of an intact acrosome, and non-fluorescent events indicate cells with a disrupted acrosome. Each sperm sample was diluted to 50 × 10^6^ cells per ml, divided into two parts and incubated either with 100 µg LPC (Sigma) or without it for 15 min in a 5% CO_2_ atmosphere at 38 °C to induce an acrosome reaction in the capacitated spermatozoa. The cells were then fixed and permeabilised with 2% paraformaldehyde in PBS (Santa Cruz Biotechnology, Dallas, TX, USA) supplemented with 0.2% (wt/vol) Triton X-100 (Sigma). After 10 min, lectin peanut PNA (Thermo Fisher Scientific) was added to sperm suspensions (2 µg/mL) to visualise acrosomes and then diluted with PBS to achieve a concentration of 300 cells/µL, placed in the 96-well plastic F-bottom clear plate and analysed in the Guava easyCyte 8HT Benchtop Flow Cytometer with the InCyte guavaSoft 4.0 software for data acquisition and analysis. The fluorescence of PNA-treated samples was excited with a 642 nm red laser, and the emitted fluorescence was captured by means of a 661/15 nm RED-R filter set. Spermatozoa were discriminated from debris with the use of a combination of forward scatter (FSC) and side scatter (SSC) signals, and 5000 cells were acquired per analysis. 

### 4.8. Evaluation of Tyrosine Phosphorylation 

Sperm samples were washed 3 times by centrifugation (350× *g* for 10 min at room temperature) with a PBS buffer supplemented with 1 mM sodium orthovanadate, which served as a tyrosine phosphatase inhibitor. Sperm pellets were suspended in Laemmli buffer, boiled for 10 min and centrifuged at 14,000× *g* for 10 min to remove cell debris. Electrophoresis SDS-PAGE was carried out using 12% Mini-PROTEAN TGX Stain-Free Precast Gels (Bio-Rad, Hercules, CA, USA). The stain-free gels were activated for 45 s and scanned in the ChemiDoc Touch Imaging System (Bio-Rad). Proteins from the gel were then electrotransferred to a nitrocellulose membrane and scanned again to obtain a stain-free image of all the transferred proteins. The membranes were blocked overnight in a cold room with a 2% skim milk dissolved in PBST (PBS with 0.1% Tween-20). Tyrosine phosphorylation of the proteins was detected using primary antibodies 4G10 Platinum (Sigma-Aldrich, St. Louis, MO, USA) diluted 1:2000 in PBST and HRP-conjugated Immun-Star anti-mouse secondary antibodies diluted 1:25,000 in PBST (Bio-Rad) and visualized using Clarity Western ECL Substrate (Bio-Rad). The stain-free blot image of total proteins, captured after a Western blot, was used as a loading control. 

### 4.9. Protein Extraction and Concentration Measurement

The aliquots containing 100 × 10^6^ of sperm cells were taken from capacitation-supporting medium at 0, 2, 4 and 6 h. After centrifugation at 900× *g*/10 min/4 °C, the sperm pellets were washed once with PBS and centrifuged at 900× *g*/10 min/4 °C. Subsequently, pellets were resuspended in 100 µL lysis buffer (7 M urea, 2 M thiourea, 4% [wt/v] 3-[(3-cholamidopropyl)-dimethylammonio]-1-propanesulfonate [CHAPS]), 2.5% (*v/v*) protease inhibitor cocktail, 1% (*v/v*) phosphate inhibitor coctail, 0.1 mM neocuproine). Additional samples were resuspended in lysis buffer with 40 mM dithiothreitol (DTT) and served as positive controls. The samples were sonicated on ice three times for 7 s at a 30% amplitude using a VCX-130 Ultrasonic Processor (Sonics & Materials, Inc., Newtown, CT, USA) and subsequently centrifuged at 1000× *g* for 30 min at 18 °C. To inhibit the thiol-disulphide exchange reaction, samples were precipitated with 350 µL of 10% trichloroacetic acid (TCA). Following this step, pellets were washed twice with cold acetone to remove acid. Prior to protein concentration measurement, samples were centrifuged at 14,000× *g*/5 min/4 °C. Next, pellets were resuspended in rehydration buffer (7 M urea, 2 M thiourea, 4% CHAPS). The protein concentration was measured using the Pierce 660 nm Protein Assay (Thermo Fisher Scientific, Waltham, MA, USA). 

### 4.10. Identification of Protein Thiol Redox State

A Saturn-2D REDOX Labelling Kit (DyeAGNOSTICS, Halle, Germany) was used for the blocking and labelling of oxidised cysteines. Five micrograms of each protein sample was mixed with 4 µL of redox labelling buffer and 3 µL of cysteine interacting compound (CinC) to block reduced cysteines. The samples were mixed, spun down briefly and incubated at 35 °C for 1h. Then, 3 µL of redox stop solution was added. The samples were then incubated at 35 °C for 10 min. The removal of excessive CinC was performed with spin columns filled with the redox matrix. The oxidised cysteines were reduced by the addition of 2.5 µL tris(2-carboxyethyl)phosphine (TCEP). The samples were mixed, spun down briefly and incubated at 35 °C for 1 h. Oxidised cysteines were labelled by adding 5 µL S-Dye300. Samples were mixed, spun down briefly and incubated at 35 °C for 1h. Then, 6 µL of redox stop solution was added to each sample, which was mixed, spun down briefly and incubated at 35 °C for 10 min. The analogous labelling of the pool of all samples from the experiment with a maleimide-based dye (S-Dye200) constituted an internal protein standard (IS). Each sample was mixed with the same amount of the internal standard and run simultaneously on a single gel. During gel analysis, IS served as a reference and allowed further easy gel-to-gel normalisation. 

### 4.11. D SDS-PAGE Separation of Fluorescently Labelled Proteins

Samples taken from capacitation-supporting medium at 0, 2, 4 and 6 h were separated by 1D SDS-PAGE to evaluate the overall level of protein-reversible oxPTMs and select the samples with the largest changes in the level of oxPTMs. Five microlitres of each sample (labelled as described in Section 4.10) was mixed with 5 μL of internal standard. Then, 5 μL of 2× Laemmli loading buffer, containing 80 mM DTT as a reductant, was added. Subsequently, 7 μL of each pooled sample containing 1 μg of protein was loaded in each single lane. Negative controls were performed without a reduction step with TCEP, and positive controls were treated with DTT after protein extraction. The proteins were resolved in 12.5% polyacrylamide gels during a standard SDS-PAGE electrophoresis. 

### 4.12. D-PAGE Separation of Fluorescently Labelled Proteins

The samples for 2D-PAGE were selected based on the SDS-PAGE results (with capacitation times of 0 h and 6 h). Five micrograms of each protein sample selected for 2D-PAGE separation (labelled as described in Section 4.10) was mixed with 5 µg of the internal standard proteins and complemented with the rehydration buffer (7 M urea, 2 M thiourea, 4% CHAPS) to the volume of 450 µL. Samples were loaded onto 24 cm Immobiline DryStrips, with a nonlinear 3−10 pH gradient (GE Healthcare, Chicago, IL, USA), and rehydrated for 12 h. Isoelectric focusing was performed as previously described [67]. Furthermore, the strips were equilibrated in SDS equilibration buffer (6 M urea, 75 mM Tris-HCl [pH 8.8], 29.3% glycerol, 2% SDS, and a trace of bromophenol blue) containing 10 mg/mL DTT for 15 min and then in SDS equilibration buffer containing 25 mg/mL iodoacetamide for 15 min. The second dimension of 2D-PAGE was performed for 16 h at 1 W/gel using an Ettan Dalt-Six apparatus (GE Healthcare). 

### 4.13. Image Acquisition and Analysis

After electrophoresis, the gels were scanned with a Typhoon 9500 FLA scanner (GE Healthcare) at excitation/emission wavelengths of 532/576 nm (S-Dye200) and 633/664 nm (S-Dye300), respectively. SameSpots software (TotalLab, Newcastle upon Tyne, England, UK) was used to detect, normalise and quantify the spot patterns of all SDS-PAGE and 2D-PAGE gels. A Student’s *t*-test analysis embedded in the software was used to detect differences in the degree of cysteine oxidation of the proteins manifested in the differences in spot fluorescence in relation to the internal standard. Proteins for which the intensity changed (*p* < 0.05, fold change >2) were selected for further mass spectrometry identification. To properly select and identify the spots, additional preparative gels were run using 200 μg of pooled proteins. Proteins were localised and identified as previously described [67].

### 4.14. Mass Spectrometry Protein Identification 

Protein spots selected as described in Section 4.13 were manually excised and subjected to trypsin digestion and spectrometric identification. Gel pieces were de-stained and washed [68]. A modified sequencing grade trypsin (Promega, Madison, WI, USA) solution (0.2 mg/mL in 50 mM ammonium bicarbonate) was used for the in-gel digestion of proteins conducted overnight at 37 °C. The gel pieces were then sonicated for 10 min to effectively release the peptides from the polyacrylamide gel, before being dried in a vacuum centrifuge. The peptides were then dissolved in 2 μL of a matrix solution prepared by suspending 5 mg of α-cyano-4-hydroxy-cinnamic acid (BrukerDaltonics, Bremen, Germany) in 1 mL of 70% acetonitrile solution with 0.1% trifluoroacetic acid. The peptides were loaded on the MT 34 Target Plate Ground Steel (Bruker Daltonics) and analysed with a MALDI Autoflex Speed TOF/TOF mass spectrometer (Bruker Daltonics) as previously described [62].

### 4.15. Gene Ontology Analysis

The UniProtKB database (http://www.uniprot.org, accessed on 2 October 2020) was used to associate the identified proteins’ accession numbers with the corresponding gene names. The potential protein–protein interactions of the identified proteins, as well as their localisation and functions, were assigned on the basis of the results generated by the Search Tool for the Retrieval of Interacting Genes (STRING) software (https://string-db.org, accessed on 7 October 2020).

### 4.16. Statistical Analyses

The results are presented as the mean ± standard error of the mean (SEM) (*n* = 8). All analyses were performed with GraphPad Prism software v. 6.02 (GraphPad Software Inc. San Diego, CA, USA) at a significance level of *p* < 0.05. The data from motility measurements and flow cytometry examinations of spermatozoa were tested for normality (Shapiro–Wilk test) and homogeneity of variance (Fisher’s F test), then analysed using one-way ANOVA, followed by Fisher’s LSD test for the post hoc comparison of means. To analyse 1D and 2D gels, the statistical component of the SameSpots software was used as described above (Section 4.13).

## 5. Conclusions

Our research provides evidence that redox modifications of proteins accompany bull sperm capacitation. This research proves that reversible oxPTMs concern proteins associated directly with capacitation and acrosome reaction at all levels from the induction of redox signalling via SOD and DLD, through transferring the oxidation signal via GAPDHS and GST and ending with the key target proteins essential for capacitation, such as ROPN1L and ACRBP (Figure 9A). Redox modifications of proteins during capacitation appear to be associated with the regulation of acrosomal, cytoskeletal and mitochondrial proteins (Figure 9B). Moreover, the redox-modified proteins found in numerous proteoforms indicate crosstalk between oxPTMs and other post-translational modifications.

The results of our study show the directions of future experiments which should aim to unravel the detailed mechanisms of the influence of redox modification on the function of specific proteins during sperm capacitation and the acrosome reaction to provide comprehensive knowledge about these processes.

## Figures and Tables

**Figure 1 ijms-22-07903-f001:**
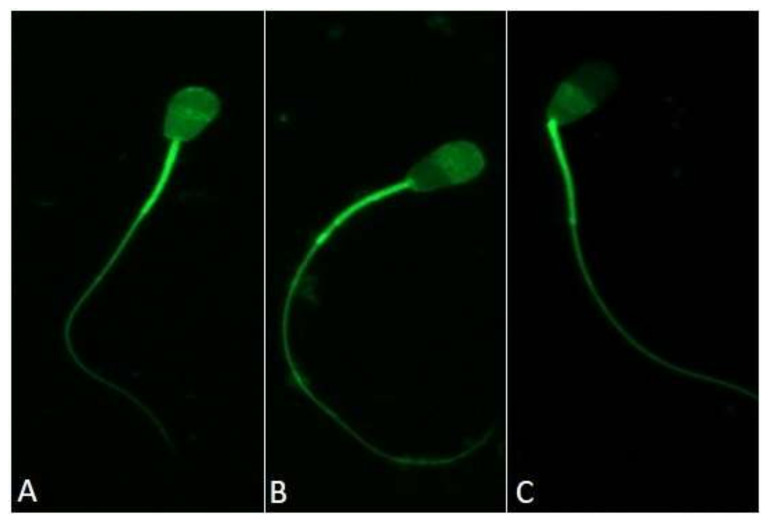
CTC fluorescence patterns of the sperm head in different stages of the capacitation process. (**A**), non-capacitated—bright fluorescence of the entire sperm head; (**B**), capacitated—prominent fluorescent-positive acrosome and fluorescence-free band in the post-acrosomal region; (**C**), acrosome-reacted sperm—minimal fluorescence signal in the sperm head with a bright signal in the equatorial segment.

**Figure 2 ijms-22-07903-f002:**
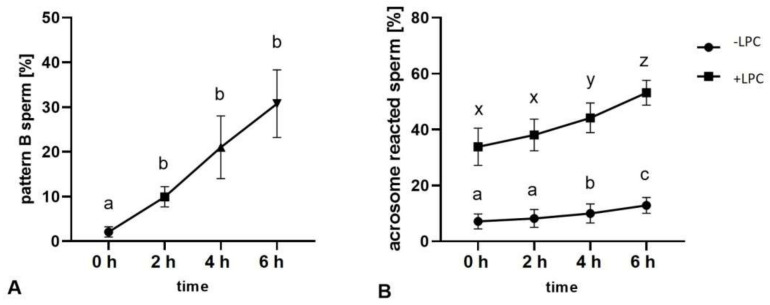
Kinetics of pattern B accumulation in the CTC-stained spermatozoa, corresponding to the progression of bull sperm capacitation (**A**); assessment of spontaneous AR (•) and LPC-induced acrosome reaction (▪) (**B**) after 0, 2, 4 and 6 h of incubation in the capacitation-supporting buffer. Different letters indicate significant differences (*p* < 0.05) among timepoints. All values are represented as the mean ± SEM (*n* = 8).

**Figure 3 ijms-22-07903-f003:**
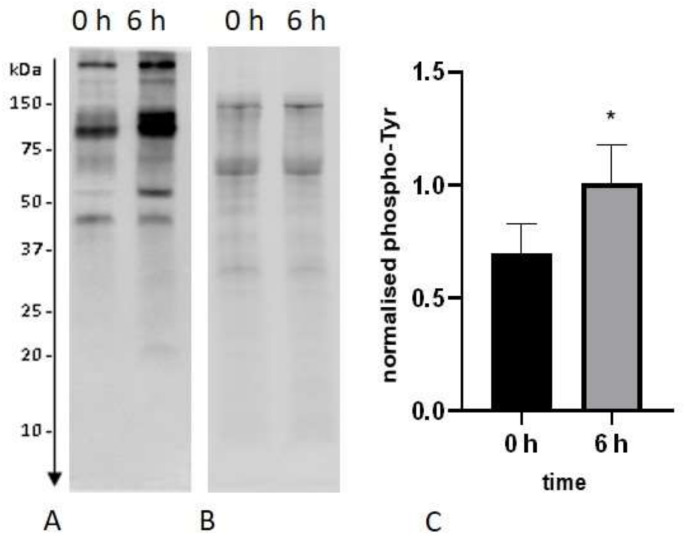
Western blot showing protein tyrosine phosphorylation after 0 and 6 h of incubation under capacitation-supporting conditions (**A**), stain-free image of total bull sperm proteins transferred to the membrane (**B**), protein tyrosine phosphorylation level after 0 and 6 h of incubation under capacitation-supporting conditions normalised to total proteins transferred to the membrane (**C**). * *p* < 0.05. Data are expressed as the mean ± SEM (*n* = 8).

**Figure 4 ijms-22-07903-f004:**
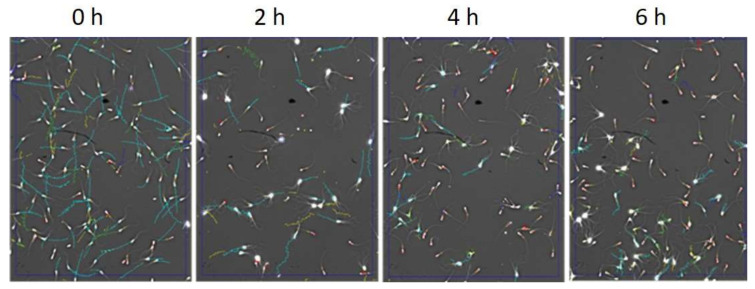
Time-lapse photos showing the degree of agglutination of the bull sperm after 0, 2, 4 and 6 h of incubation in capacitation-supporting buffer.

**Figure 5 ijms-22-07903-f005:**
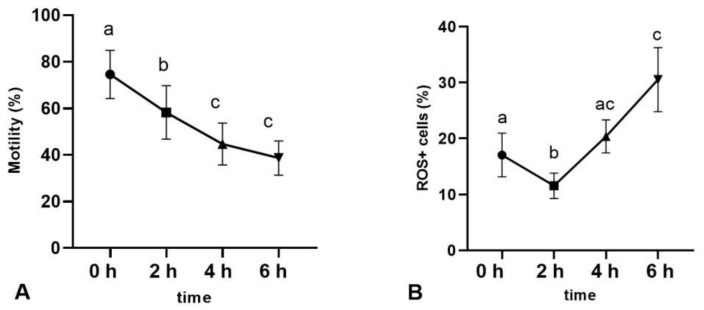
Total sperm motility (%) (**A**) and ROS-positive spermatozoa content (%) (**B**) after 0, 2, 4 and 6 h of incubation in capacitation-supporting conditions. Different letters indicate significant differences (*p* < 0.05) among timepoints. Data are expressed as the mean ± SEM (*n* = 8).

**Figure 6 ijms-22-07903-f006:**
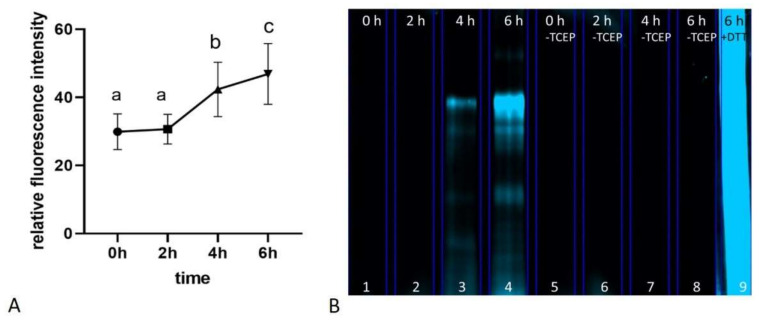
Relative fluorescence intensity corresponding to the levels of reversible oxPTMs of bull sperm incubated for 0, 2, 4 and 6 h under capacitation-supporting conditions. Different letters indicate significant differences (*p* < 0.05) among timepoints. Data are expressed as the mean ± SEM (*n* = 8) (**A**). Representative polyacrylamide gel showing reversible protein oxPTMs of bull sperm incubated for 0, 2, 4 and 6 h under capacitation-supporting conditions (lines 1–4), negative controls without a protein reduction step with TCEP (lines 5–8) and positive control treated with DTT after protein extraction (line 9) (**B**).

**Figure 7 ijms-22-07903-f007:**
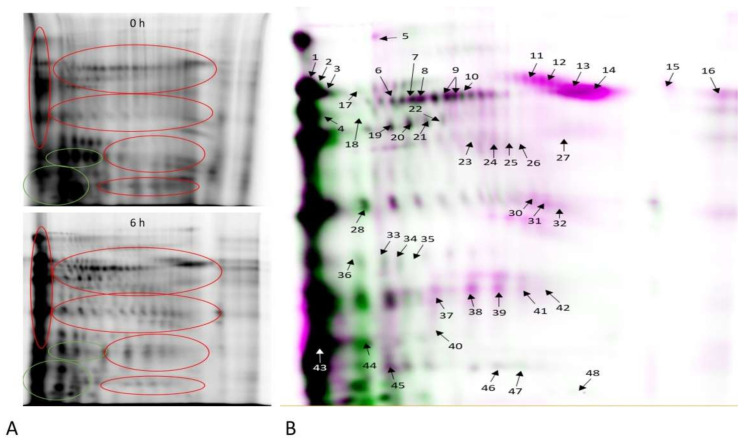
Representative 2D gel images of proteins showing redox changes between 0 and 6 h of incubation in capacitation-supporting conditions (*n* = 8). Proteins with reduced cysteines after 6 h of capacitation are marked in green. Proteins with oxidised cysteines after 6 h of capacitation are marked in red (**A**). Representative 2D gel showing redox-modified proteins after 6 h of incubation in capacitation-supporting conditions (**B**). Redox-modified protein spots are listed in Table 1.

**Figure 8 ijms-22-07903-f008:**
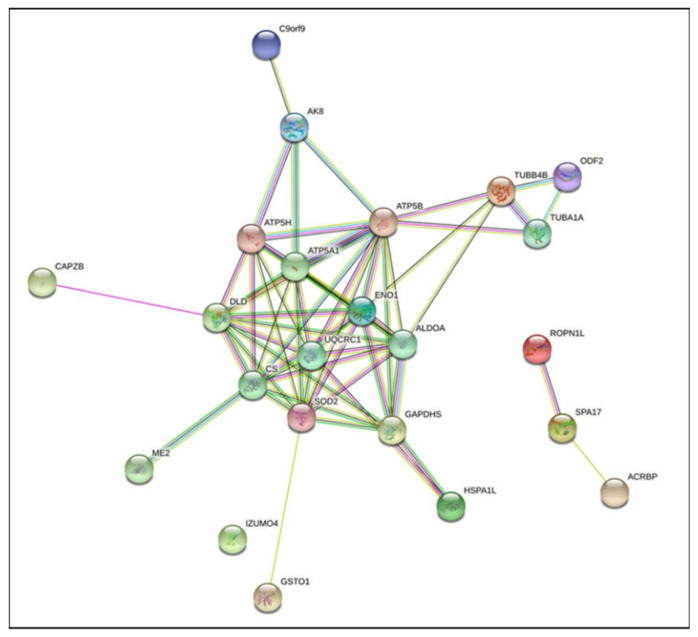
STRING protein association analysis of proteins showing changes in oxPTM levels after 6 h of incubation under capacitation-supporting conditions. Proteins are listed and characterised in Table 1. Network nodes represent proteins, and the connecting lines represent protein–protein interactions.

**Figure 9 ijms-22-07903-f009:**
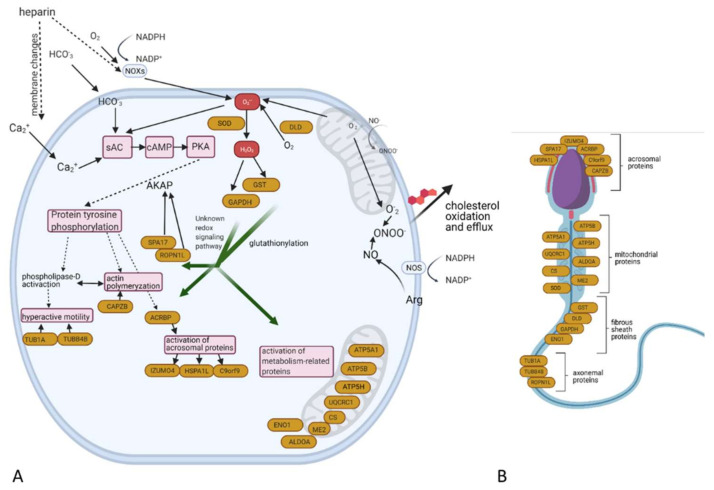
The proposed scheme for redox signal transduction during the capacitation of bull spermatozoa (**A**). Identified redox-modified proteins are highlighted by yellow ovals. Generation of O_2_^-^ in spermatozoa is catalysed mainly via NADPH oxidases, mitochondria (which generate a low level of ROS during steady-state respiration) or DLD. This O_2_^−•^ combines with •NO produced by nitric oxide synthase (NOS), resulting in the generation of the oxidant ONOO^−^, which is responsible for the oxidation of cholesterol to oxysterols, which dramatically increase membrane fluidity. The combination of O_2_^−•^, HCO_3_^−^ and Ca^2+^ activates soluble adenylyl cyclase (sAC), which stimulates cAMP production and the further activation of protein kinase A (PKA), leading to the tyrosine phosphorylation of proteins. The combined action of H_2_O_2_ (generated as a result of SOD activity) and NO (generated by NOS) triggers the transmission of the redox signal via the activity of GST and GAPDHS to target proteins related directly to tyrosine phosphorylation, actin polymerisation, the activation of acrosomal proteins, hyperactivation of motility and energetic metabolism (see Table 1). A detailed description of this can be found in the text. Distribution of redox regulated proteins in sperm structures (**B**).

**Table 1 ijms-22-07903-t001:** Proteins showing changes in oxPTM levels after 6 h of incubation in capacitation-supporting conditions; oxidised (ox) or reduced (red) cysteines (*n* = 8 in each group; *p* < 0.05).

Spot No. in Figure 7	Cys Redox Modification	Protein Name (Gene Name)	Accession Number	Calculated MW/pI	Protein Score
1	ox	Tubulin beta-4B chain *(TUBB4B)*	Q3MHM5	50,255/4.79	533/5
2	ox	Tubulin beta-4B chain *(TUBB4B)*	Q3MHM5	50,255/4.79	605/7
3	ox	Tubulin alpha-3 chain *(TUB1A)*	Q32KN8	50,578/4.97	702/7
4	ox	ATP synthase subunit beta, mitochondrial *(ATP5B)*	P00829	56,249/5.15	490/4
5	ox	Heat shock 70 kDa protein 1-like *(HSPA1L)*	P0CB32	70,744/5.89	112/2
6	ox	ATP synthase subunit alpha, mitochondrial *(ATP5A1)*	P19483	59,797/9.21	529/5
7	ox	ATP synthase subunit alpha *(ATP5A1)*	F1MLB8	59,767/9.21	301/2
8	ox	ATP synthase subunit alpha *(ATP5A1)*	F1MLB8	59,767/9.21	427/4
9	ox	ATP synthase subunit alpha *(ATP5A1)*	F1MLB8	59,767/9.21	259/3
10	ox	Dihydrolipoyl dehydrogenase *(DLD)*	F1N206	54,723/7.59	84/2
11	ox	Glyceraldehyde-3-phosphate dehydrogenase, testis-specific *(GAPDHS)*	Q2KJE5	43,659/8.32	217/3
12	ox	Glyceraldehyde-3-phosphate dehydrogenase, testis-specific *(GAPDHS)*	Q2KJE5	43,659/8.32	421/3
13	ox	Glyceraldehyde-3-phosphate dehydrogenase, testis-specific *(GAPDHS)*	Q2KJE5	43,659/8.32	368/3
14	ox	Glyceraldehyde-3-phosphate dehydrogenase, testis-specific *(GAPDHS)*	Q2KJE5	43,659/8.32	35/3
15	ox	Glyceraldehyde-3-phosphate dehydrogenase, testis-specific *(GAPDHS)*	Q2KJE5	43,659/8.32	347/3
16	ox	Glyceraldehyde-3-phosphate dehydrogenase, testis-specific *(GAPDHS)*	Q2KJE5	43,659/8.32	345/3
17	ox	Outer dense fibre protein 2 *(ODF2)*	Q2T9U2	76,249/7.52	212/3
18	ox	Cytochrome b-c1 complex subunit 1, mitochondrial *(UQCRC1)*	P31800	53,444/5.94	146/2
19	ox	2-phospho-D-glycerate hydro-lyase *(ENO1)*	A0A3Q1LGA6	45,640/7.01	172/2
20	ox	2-phospho-D-glycerate hydro-lyase *(ENO1)*	A0A3Q1LGA6	45,640/7.01	159/2
21	ox	2-phospho-D-glycerate hydro-lyase *(ENO1)*	A0A3Q1LGA6	45,640/7.01	125/2
22	ox	2-phospho-D-glycerate hydro-lyase *(ENO1)*	A0A3Q1MXQ0	54,774/9.12	299/2
23	ox	Citrate synthase, mitochondrial *(CS)*	Q29RK1	51,968/8.16	85/2
24	ox	Citrate synthase, mitochondrial *(CS)*	Q29RK1	51,968/8.16	64/2
25	ox	Fructose-bisphosphate aldolase *(ALDOA)*	A6QLL8	39,925/8.45	91/2
26	ox	Fructose-bisphosphate aldolase *(ALDOA)*	A6QLL8	39,925/8.45	218/2
27	ox	Fructose-bisphosphate aldolase *(ALDOA)*	A6QLL8	39,925/8.45	367/5
28	ox	F-actin-capping protein subunit beta *(CAPZB)*	A0A3Q1LPZ4	31,439/5.74	124/2
29	ox	F-actin-capping protein subunit beta *(CAPZB)*	A0A3Q1LPZ4	31,439/5.74	150/2
30	ox	Malate dehydrogenase *(ME2)*	A0A3Q1LTB0	33,440/8.83	129/2
31	ox	Malate dehydrogenase *(ME2)*	A0A3Q1LTB0	33,440/8.83	235/3
32	ox	Malate dehydrogenase *(ME2)*	A0A3Q1LTB0	33,440/8.83	359/3
33	red	IZUMO family member 4 *(IZUMO4)*	A0A3Q1MGI1	16,918/5.62	178/2
34	red	IZUMO family member 4 *(IZUMO4)*	F1MYA6	24,986/6.24	135/2
35	red	IZUMO family member 4 *(IZUMO4)*	F1MYA6	24,986/6.24	136/2
36	red	Acrosin-binding protein *(ACRBP)*	E1B7S8	62,374/4.99	248/3
37	ox	IZUMO family member 4 *(IZUMO4)*	F1MYA6	24,986/6.24	170/2
38	ox	IZUMO family member 4 *(IZUMO4)*	F1MYA6	24,986/6.24	165/3
39	ox	IZUMO family member 4 *(IZUMO4)*	F1MYA6	24,986/6.24	121/2
40	ox	Glutathione S-transferase *(GST)*	Q2KIV8	27,174/6.83	171/2
41	ox	Glyceraldehyde-3-phosphate dehydrogenase, testis-specific *(GAPDGS)*	Q2KJE5	43,659/8.32	132/2
42	ox	Glyceraldehyde-3-phosphate dehydrogenase, testis-specific *(GAPDHS)*	Q2KJE5	43,659/8.32	140/2
43	ox	Sperm autoantigenic protein 17 *(SPA17)*	F1MI43	16,949/4.88	73/2
44	red	Ropporin-1-like protein *(ROPN1L)*	Q3T064	24,753/6.10 24,125/5.22	185/4 142/2
45	red	ATP synthase subunit d, mitochondrial *(ATP5H)*	P13620	18,738/5.99	308/3
46	red	Superoxide dismutase *(SOD)*	E1BHL1	52,348/10.37	73/2
47	red	Superoxide dismutase *(SOD)*	E1BHL1	52,348/10.37	72/2
48	red	Sperm acrosome associated 9 *(C9orf9)*	A0A3Q1MYU9	25,520/8.96	549/5

## Data Availability

All data underlying this article are available directly in the article text.

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
