# Peer review of "Bull Sperm Capacitation Is Accompanied by Redox Modifications of Proteins"

_ijms, 2021, doi:10.3390/ijms22157903_

Round 1

Reviewer 1 Report

This is a review on the article “Bull sperm capacitation is accompanied with redox modifications of proteins”, currently submitted to the International Journal of Molecular Sciences for potential publication.

The authors describe the accumulation of ROS during the capacitation of bull sperm as well as proteins that are presumably modified by oxidative processes. For their investigations, the authors used different methods, such as fluorescence microscopy, flow cytometry, Western Blotting and proteomic approaches, including mass spectrometry.

My major concerns are:

I could not find any ethics statement. However, using animal samples this is mandatory.

PBS was used to wash the sperm (lane 689), even though the authors wanted to show protein phosphorylation. This is not understandable because for other experiments HEPES or TBS was used. Phosphate ions from the PBS buffer might lead to interferences in experiments investigating phosphorylation. Therefore, PBS should not be used in any preparation step. The authors should provide results without the use of PBS.

Did the authors prove in advance that the semen diluent really was protein-free, for example by Western Blot? Furthermore, Bioxcel semen diluent contains four antibiotics. How can the authors be sure that there were no bacteria in the ejaculates prior to the dilution? Antibiotics were shown to induce ROS production in bacteria leading to their death increasing the ROS concentration in samples.

Here are my other comments:

Please change “with” to “by” in the title

Introduction:

In lane 56 there is probably a mistake. The reaction of NO and dioxide (superoxide) does not result in NO + ONOO- but in peroxynitrite (ONOO-) only. Please correct.

Please always write ions and radicals with a superscript – or + or •. Please correct throughout the manuscript. It is “Ca2+” (also throughout the text!).

Lane 63: It should be “Sperm mitochondria are an important…”

Please rephrase the sentence in lane 65-67. It is misleading to say that mitochondria support sperm capacitation.

Materials:

Why were the sperm incubated at 38°C and not at 37°C?

Lane 581: It should be MgCl2 × 4 H2O.

Lane 691: It should be “glycerol”.

The affiliation of a company needs to be given only once (e.g., lanes 673, 697, 703, 704)

Lane 709: Please write 100 × 108 cells.

Please always use a comma to separate thousands (e.g., lanes 720, 724, 775, 776)

Even though TBST (lane 700) probably is TBS containing Tween-20, this should be explained once.

Please describe the mass spectrometric approach in more detail and provide structures of the cleavages mass spectra

Results:

The authors write about significant differences. Where are the significances in the graphs? Are the lower case letters supposed to show significances? But why is there a letter at the time point 0 h? Anyway, if the significances are depicted by the letters, this needs to be mentioned in the figure legends. Furthermore, letters are not good, please use asterisks.

How do the authors know that the “largest accumulation” is observed after 6 h? Where there also time points after 6 h? If so, these should be depicted. If not, the sentence (lane 121) should be rephrased.

Page 4 is completely empty.

The data shown in the graphs should be given as mean ± standard deviation and not SEM. SEM obfuscates the true biological range.

In Figure 2 there is no necessity to use different symbols for the different time points. Please just use one symbol. As already stated above, what are the lower case letters for? The same is true for Figure 5 and 6A.

In Figure 3A, what is the arrow at the kDa scale for? Is it to show the direction of migration? This is not necessary. What is shown in the bar graph in Figure 2C? Mean ± SD or SEM? Please add the information.

Figure 4 is not clear. There are blue, red and green things visible. However, the figures are so small that it is not possible to see anything. Either enlarge the figure or delete it completely.

Also the Figures 7 - 9 are too small. As IJMS is an online journal, there is no need to make the Figures this small.

Discussion:

Lane 237: “… is has been shown…”

Lane 240: “I our approach the sperm…”

Lane 242: “A gel-based proteomic approach…”

Lane 268: There is something missing at the end of the title

Lane 277: See text. According to lane 166 and Fig. 5B ROS production exceeds 30% after 6 h.

Lane 282-283: What does it mean the exposure is short-lived? Please rephrase the sentence.

Is it GAPDH or GAPDHS (spermatogenic GAPDH) (Figure 9 and text)?

Lane 415: Again, please use the right spelling for superoxide and nitric oxide.

Lane 416: “…which catalyses…”

Lane 850: “…during modification appear to be…”

References:

in the reference list should be given consistently and as follows: 8. Bowman, C.M.; Landee, F.A.; Reslock, M.A. Chemically Oriented Storage and Retrieval System. 1. Storage and Verification of Structural Information. J. Che

Reviewer 2 Report

The manuscript entitled “Bull sperm capacitation is accompanied with redox modifications of proteins” by Mostek et al. is a research article on the mechanism of bovine sperm capacitation. The results of this study demonstrated that sperm capacitation is accompanied with redox modifications of 22 proteins involved in the production of ROS, in downstream redox signal transfer, in the cAMP/PKA pathway, acrosome exocytosis, actin polymerisation and hyperactivation. The manuscript provides interesting information for the mechanism of sperm capacitation; it is well written and in good English, concluded that the manuscript is appropriate for publication. Authors should address the following issues:

- p.10, line 673: Is Triton X-100 at a concentration of 0.2% safe enough to induce only permeabilization and not sperm extraction?

- p.10, line 596: The operator should note the degree of sperm agglutination and the type of sperm union (e.g., grade 2, head-head unions).

Round 2

Reviewer 1 Report

Please always write Ca2+ and not 2+ in subscript (lane 58).

Please correct lane 658/659. Was it TBST or PBST?

Author Response

Please always write Ca2+ and not 2+ in subscript (lane 58).

Corrected

Please correct lane 658/659. Was it TBST or PBST

It was PBST, Corrected